# Highly Efficient No-reference 4K Video Quality Assessment with Full-Pixel Covering Sampling and Training Strategy

## ABSTRACT

Deep Video Quality Assessment (VQA) methods have shown impressive high-performance capabilities. Notably, no-reference (NR) VQA methods play a vital role in situations where obtaining reference videos is restricted or not feasible. Nevertheless, as more streaming videos are being created in ultra-high definition (*e.g.*, 4K) to enrich viewers' experiences, the current deep VQA methods face unacceptable computational costs. Furthermore, the resizing, cropping, and local sampling techniques employed in these methods can compromise the details and content of original 4K videos, thereby negatively impacting quality assessment. In this paper, we propose a highly efficient and novel NR 4K VQA technology. Specifically, first, a novel data sampling and training strategy is proposed to tackle the problem of excessive resolution. This strategy allows the VQA Swin Transformer-based model to effectively train and make inferences using the full data of 4K videos on standard consumer-grade GPUs without compromising content or details. Second, a weighting and scoring scheme is developed to mimic the human subjective perception mode, which is achieved by considering the distinct impact of each sub-region within a 4K frame on the overall perception. Third, we incorporate the frequency domain information of video frames to better capture the details that affect video quality, consequently further improving the model's generalizability. To our knowledge, this is the first technology for the NR 4K VQA task. Thorough empirical studies demonstrate it not only significantly outperforms existing methods on a specialized 4K VQA dataset but also achieves state-of-the-art performance across multiple open-source NR video quality datasets.

## CCS CONCEPTS

• **Computing methodologies** → **Image processing**; • **Information systems** → *Multimedia information systems*.

## KEYWORDS

4K video quality assessment, 4K video sampling strategy, Network training, State-of-the-art

## 1 INTRODUCTION

Advancements in multimedia technology have facilitated the distribution of videos across multiple platforms [19, 37]. However, the quality of these videos varies significantly. Thus, Video Quality

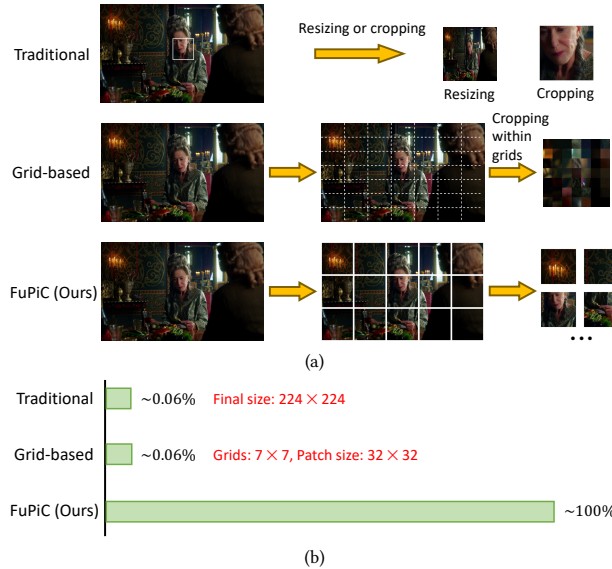

(a)

(b)

**Figure 1: Comparison of data sampling strategies in VQA methods on 4K videos. Fig.(a) visually shows the commonly used sampling strategies and ours proposed strategy. Fig.(b) illustrates the content-covering percentage of sampling strategies on 4K videos. Both traditional and grid-based strategies cover only a minimal amount of content, where the grid-based method also accesses some global information. In contrast, our strategy is capable of covering the entire content.**

Assessment (VQA) has become essential for accurately understanding and predicting the quality of user experience (QoE) [3, 43, 44]. Moreover, advances in hardware and the increasing demand for high-quality, high-resolution videos have led to a surge in 4K videos. Unfortunately, many 4K videos are not originally filmed in 4K but are instead generated from lower resolutions using techniques such as up-sampling and super-resolution [23]. This has led to massive low-quality pseudo-4K videos, notably diminishing user experience on social platforms and escalating resource usage. Hence, it is crucial to research and understand the distinctions among various 4K video types, including those from the source medium, high-quality 4K restored videos, and mid-to-low quality pseudo-4K videos. Nonetheless, despite the urgent necessity, a specific approach for no-reference (NR) 4K VQA is currently lacking.

### Challenges:

While current VQA techniques have shown considerable advancements [15, 23, 40, 44, 46], they encounter various limitations when handling NR 4K videos because of the demanding requirements in terms of high-resolution and computational resources. Typically, early deep learning methods for VQA involve traditional

pre-processing of video frames, such as resizing or cropping, before performing quality evaluation [23, 40, 46]. Actually, resizing a frame can cause the loss of crucial details, and cropping may result in the omission of essential visual information from the original 4K video. As a result, both actions have the potential to compromise the performance of VQA.

To mitigate the impact of the above operations on VQA, FAST-VQA proposes a novel data sampling strategy based on Grid Mini-patch Sampling (GMS) [41]. This data sampling strategy, named as "fragments", enables the learning of local detail information (through cropping) while retaining certain global information (as grids are uniformly divided across the entire frame). This innovative approach has been successfully applied in VQA, and numerous subsequent methods are proposed based on it [20, 43]. Among them, DOVER leverages the network of FAST-VQA and combines it with aesthetic and technical perspectives to better understand and predict the QoE on videos [43]. SAMA further improves upon FAST-VQA, enhancing its performance with minimal additional resource consumption [20]. Despite this, when cropping within a uniformly distributed grid, the grid size increases notably with higher resolutions such as 4K videos. Consequently, if the input data size to the network stays constant [41], there is a rapid decrease in the proportion of useful information acquired.

To better explain the different information retrieval rates of different sampling strategies, a comparison is made and shown in Fig. 1. It is clear to see that both traditional and grid-based strategies can only cover a tiny fraction of the content in 4K videos. As a result, the grid-based strategy almost entirely loses the global semantic information, as shown in Fig. 1(b). Moreover, in the commonly seen 4K video resolution (*i.e.*, 3840x2160), the cropped patches occupy only about 0.6% of the grid areas. This restriction significantly limits the effectiveness of 4K VQA and possibly for videos of even greater resolutions in the future.

*Motivation and Main Idea:*

High-resolution 4K videos present a challenge to existing data sampling methods in capturing adequate information. Hence, the vital issue we need to handle is ***how to maximize video information fed into the network with limited computational resources.*** To confront this core issue head-on, this paper proposes a highly efficient and novel data sampling and training strategy for NR 4K VQA named **Fu**ll-**Pi**xel **C**overing (FuPiC). This strategy ensures that the network can capture nearly all content of the sampled frames while guaranteeing an efficient training process. Generally, 4K videos are partitioned into non-overlapping patches in our study, ensuring the content can be processed within a single GPU for training. Adopting this full sampling approach, we thoughtfully develop a novel training strategy. Instead of the conventional method of treating samples within one batch as "supervisory units," we opt to gather multiple output results from samples originating from the same frame and supervise the combination of these results. Stemming from this concept, FuPiC enables the network to receive and learn from the entire content (the cropped patches occupy 100%, far exceeding 0.6%. as depicted in Fig. 1), simultaneously ensuring that GPU resources are adequate for handling the learning process of videos with high-resolution.

As stated earlier, FuPiC combines the results from different samples and supervises the combined result. However, the second challenge we need to face is ***how to effectively and reasonably combine these results.*** To tackle this challenge, inspired by the subjective perception model of humans when evaluating videos, a weighting and scoring scheme named region-aware scoring scheme is proposed to focus on different regions within the video. In subjective 4K video assessments, humans evaluate various regions, such as focal areas, background, and blur-affected regions due to jitter, each impacting overall quality differently. Thus, we consider partitioned patches as a series of regions, using a neural network to learn their weights and scores to estimate overall video quality.

FuPiC feeds all the information of 4K videos into the network, and ***whether the network can more effectively capture and utilize these details*** is the third point of concern in our study. One notable observation from various real-world 4K video samples is the difference in detail between high and low scoring videos, where higher scoring videos tend to contain more intricate information. To delve deeper into the impact of these details on 4K video quality, we employed Haar Wavelet Transform to shift video frames from the image domain to the frequency domain. We observed that videos with similar scenes tend to have richer high-frequency information if they score higher, which becomes especially apparent after applying Haar Wavelet processing. Consequently, we pre-process video frames with Haar Wavelets and use a linear embedding layer to handle information across different frequencies, integrating this data for network processing. This approach, named multi-frequency feature fusion, enhances the performance of the network in assessing 4K video quality, and notably, the inference time was reduced to 25% of the original one.

Additionally, a novel 4K VQA dataset with a large variety of scenes and a reasonable quality distributed range for targeted 4K VQA methods is constructed. To better meet the practical demands of the 4K VQA task, our dataset construction follows a NR paradigm instead of a reference-based one, meaning that all video clips originate from heterogeneous content. Specifically, our dataset encompasses 200 ten-second video clips sourced from a diverse array of occupationally generated content (OGC), including movies, television dramas, and TV shows from different eras and countries. Initially, we selected 200 long 4K videos guided by a criteria that the era, the genre, and the region of these videos should be distributed as balanced as possible. Subsequently, we randomly extract 10 ten-second short video clips from each long video, resulting in a total of 2000 clips. Following this, we employ a sampling strategy similar to [39, 44] to distill these into 200 representative ten-second video clips while some key video indicators are considered, including spatial activity, temporal activity, noise, brightness, and contrast. Then we adopt the Pair Comparison method in our subjective experiment, which can efficiently obtain an accurate mean opinion score (MOS) for each 4K video clip in the dataset.

*Contributions:*

To summarize, the main contributions of this work include:

- We propose a novel data sampling and training strategy, namely **Fu**ll-**Pi**xel **C**overing (FuPiC), for NR 4K VQA. FuPiC enables the network to receive all content information for learning, while ensuring an easy training process.

- We introduce a weighting and scoring scheme for FuPiC. This scheme emulates the subjective quality evaluation procedure, which can predict the frame score and eventually the overall video score more accurately.
- By transforming video frame patches into the frequency domain and integrating information from various frequency domains into the network, we emphasize the network's attention on the influence of high-frequency information on video quality. This approach enhances the network performance while mitigating the computational burden.
- We have constructed the first dataset explicitly tailored for NR 4K VQA.

Our approach significantly outperforms other VQA methods on our 4K VQA dataset. Additionally, experiments on other open-source VQA datasets also demonstrate the excellent performance of our method.

## 2 RELATED WORK

### 2.1 Classical VQA Methods

Classic VQA methods [6, 12, 18, 25–28, 31, 32, 36, 38] rely on hand-crafted features for quality evaluation. Among them, V-BLIINDS [32] is a spatio-temporal Natural Scene Statistics (NSS) model that quantifies the NSS features of frame differences and motion coherency characteristics to assess video quality; VIIDEO [26] embodies models that capture intrinsic statistical regularities of natural videos to quantify disturbances introduced by distortions; TLVQM [12] first computes temporal low-complexity handcrafted features and then uses them to extract high-complexity features. Moreover, CNN-TLVQM [13] integrates TLVQM with spatial features extracted from a pre-trained convolutional neural network (CNN) for VQA tasks.

### 2.2 Deep VQA Methods

Classical VQA methods are known for their inefficiency and struggle to perform well amidst the rapid development of multimedia technologies. With the advent of deep learning, there has been a significant emergence of deep VQA methods. These methods leverage the power of deep neural networks (DNN) to provide more accurate and efficient quality assessments, addressing the limitations of classical VQA techniques.

In earlier research, the majority of efforts are based on employing CNN/3D-CNN models [2, 14–16, 37, 40, 44–46] to delve into the spatial-temporal information within videos for quality assessment. Li et al.[15] utilizes a pre-trained CNN model to extract features and further expanded this approach to MDTVSFA [16] to explore the effectiveness of training a unified model across multiple datasets. Ying et al.[45] introduces a local-to-global architecture for predicting the overall video quality and leveraged the proposed PVQ Mapper to better learn spatial and temporal features. Wang et al.[40] investigates the impact of content, technical quality, and compression level on video quality. Xu et al.[44] incorporates a Graph Convolution Module into VQA to capture long-distance and cross-scale relations. Sun et al.[37] proposes an efficient network that directly extracts quality-aware features from raw pixels of video frames, while extracting the motion features for accurately prediction.

Later, as the Visual Transformer evolves, Transformer-based methods [19, 20, 41–43] begin to exhibit strong performance in VQA tasks. However, the Visual Transformer takes a high computational cost, often relying on scaling or cropping video samples to reduce the cost. To overcome the limitations associated with traditional scaling and cropping strategy, Wu et al.[41] introduces a grid-based sampling method, allowing for the simple and efficient use of Video Swin Transformer [22] for end-to-end training and fine-tuning, while maintaining a focus on global information. Building on this grid-based approach, DOVER [43] explores the influence of Aesthetics and Technical quality on subjective evaluations; SAMA [20] further refines the grid-based sampling method by incorporating scaling and masking strategies to encompass multi-scale information within regular-sized inputs. Despite the improvements brought by grid-based sampling to DNN-based methods in assessing the quality of 1080P videos, challenges persist when dealing with higher-resolution content (4K and beyond). These methods face significant limitations as they only sample a small fraction of the video's content, impacting their performance adversely.

### 2.3 Databases for Video Quality Assessment

Databases are crucial for VQA tasks. Many VQA Databases [7, 9, 29, 30, 34, 35, 39, 45] are built to tackle the challenge of VQA problems. Among them, early VQA databases [30, 34] usually consist of limited numbers of source reference sequences (SRC) and then introduce different degradations such as compression artifacts or transmission errors to generate distorted sequences. Subsequently, some in-the-wild databases [7, 9, 35, 39] have significantly promoted the development of VQA tasks, due to their large diversity of content, different levels of degradations and diverse types of distortions. However, a considerable number of the videos in these databases are of low resolution, which does not meet the urgent requirements on the evaluation of high-resolution video sequences, especially in the Quality Control (QC) of the delivery of Full-High Definition (HD) or Ultra-HD source videos. For instance, KoNViD-1k [9] consists of 540P videos. With the increasing popularity of 4K videos, some databases [4, 24] that exclusively include 4K videos are proposed. However, these databases are built by conducting compression and up-scaling on a few SRC videos, resulting in the simplicity of video scenes. In contrast, our constructed 4K dataset is built on 200 4K videos with a diverse range of scenes and quality levels.

### 2.4 Quality Assessment for 4K Images/Videos

Only a few methods [23, 47] have been specifically designed for 4K image/video quality assessment tasks due to the unique challenges posed by high-resolution content. Zhu et al.[47] proposes to select three cropped patches with size $240 \times 240$ within 4K images based on local variances for efficiently extracting features. Lu et al.[23] utilizes texture complexity measure to select three patches with size $240 \times 240$ as inputs into their proposed BTURA model for 4K content quality assessment. Moreover, BTURA is trained by utilizing an extra information that whether a 4K content is true or pseudo, which could potentially restrict the practical applications of BTURA.

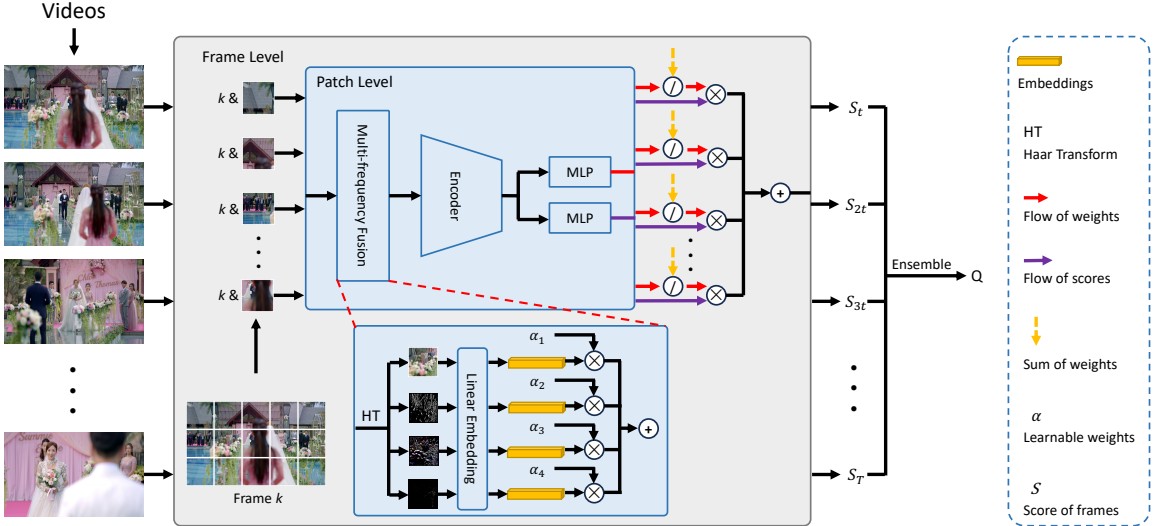

Figure 2: Overall of our proposed method. We utilize a Swin Transformer as the Encoder to extract features.

## 3 PROPOSED METHOD

The pipeline of our proposed method is illustrated in Fig. 2. As shown in the figure, the Full-Pixel Covering (FuPiC) sampling and training strategy is implemented on the frames extracted from the 4K video. The grey area that outputs the score for each frame represents FuPiC. In this process, the Swin Transformer is equipped as the encoder to extract the features for the input frames, and a proposed region-aware scoring scheme is utilized to predict the frame score based on the patch score. After FuPiC, the scores of all extracted frames are aggregated to compute the final score for the video. Moreover, we use multi-frequency feature fusion to improve the performance of our method. In this section, we first briefly introduce the Swin Transformer in Sec. 3.1. Following that, we introduce FuPiC in detail (Sec. 3.2). Then we present the region-aware scoring scheme in Sec. 3.3. The multi-frequency feature fusion is described in Sec. 3.4. Finally, we introduce the dataset we constructed for the NR 4K VQA task in Sec. 3.5.

### 3.1 Brief Introduction of Swin Transformer

The Visual Transformer [5] has achieved significant success in the field of computer vision, including image/video quality assessment [11, 19, 20, 41–43]. However, the Visual Transformer conducts self-attention across the entire content, leading to excessive computational burden. In contrast, the Swin Transformer [21] performs self-attention within windows and employs shift window operations to ensure that self-attention covers the entire content. This method significantly reduces computational complexity while maintaining model effectiveness, enabling GPUs to handle larger pixel content as input. Therefore, our method utilizes the Swin Transformer as the encoder to extract features from video frames.

### 3.2 FuPiC Sampling and Training Strategy

A basic Swin Transformer is typically trained on training samples of $224 \times 224$ or $384 \times 384$ pixels due to the limitations of Windows and

the computational capabilities of standard GPUs.[21]. Even when opting for the larger size of $384 \times 384$ pixels, it remains significantly smaller than the resolution of 4K videos. To tackle the issue, the **sampling** strategy in FuPiC partitions each frame of 4K video into non-overlapping patches with the size of $384 \times 384$ pixels as the training samples. Compared with resizing the 4K frame into the size of $384 \times 384$ pixels, our proposed sampling strategy can retain the full details of the 4K video as illustrated in Fig. 3.

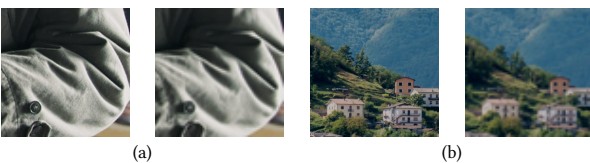

(a)                              (b)

Figure 3: Comparison of the same region in FuPiC and resizing strategies. On the left side of each example is the patch with the size of $384 \times 384$ pixels sampled by FuPiC, while on the right side is the corresponding region after resizing the 4K frame to $384 \times 384$ pixels. Details can be seen more clearly when zoomed in.

Following the sampling strategy, the patches with the size of $384 \times 384$ can be inputted into the network to achieve training or inference. However, the network training process typically supervises individual samples within one batch, treating each sample as a "supervision unit". This leads to a situation where each patch must be assigned a score for training. Nevertheless, since each patch represents only a portion of a video frame, we cannot assign the same score of the video to each patch. Therefore, the **training** strategy of FuPiC is proposed to supervise the subset of training samples in one batch rather than each individual training sample. Specifically, the training strategy of FuPiC arranges for all the patches (samples) from the same 4K frame to be in the same training batch, and the frame ID of each patch is also inputted. In each training iteration,

the outputs for patches(training samples) are transformed into the outputs for frame (subsets of samples) by Eq. 1.

$$O_f^k = \sum_{i=1}^{N} o_i^k / N, \tag{1}$$

where $o_i^k$ represents the network output of each sample from the same frame, $k$ is the frame ID, $N$ indicates the number of partitioned patches and $O_f^k$ denotes the predicted score for frame $k$. Subsequently, $O_f^k$ can be supervised by the subjective score $S_f^k$ of frame $k$, utilizing Mean Squared Error (MSE) loss as defined by the equation:

$$\mathcal{L} = \|O_f^k - S_f^k\|_2^2. \tag{2}$$

To summarize, the proposed FuPiC sampling and training strategy can input all the original content of the 4K video frame in the form of multiple patches while treating a complete 4K frame as a "supervision unit".

## 3.3 Region-aware Scoring Scheme

Utilizing FuPiC sampling and training strategy, our method ensures the entirety of 4K video content can be fed into the network for training, and each frame is treated as a "supervision unit". However, Eq. 1 assumes that each sample from this unit impacts the unit equally. This assumption introduces a challenge, as different samples (patches) from the same frame may not uniformly influence the quality of the frame. Consequently, it becomes imperative to devise a scheme for aggregating the scores of individual samples, obtained through parallel network outputs, into a final score for the frame.

To tackle this challenge, region-aware scoring scheme is proposed to consider the samples as a series of regions within the entire frame. Subjective quality scores provided by viewers are often influenced by distinct regions within the video, such as areas of focus, background, zones impacted by motion blur, and regions with intricate details. To mimic the varied effects that different regions may exhibit, the network outputs weights $w_i^k$ for all parallel input patches, reflecting their potential influence on the frame's quality. Once these weights are computed, the contribution of each region to the overall quality $y_j^k$ and region-aware score of frame $k$ can be calculated as:

$$y_j^k = w_j^k / \sum_{i=1}^{N} w_i^k, \tag{3}$$

$$O_f^k = \sum_{i=1}^{N} y_i^k \times o_i^k. \tag{4}$$

With the region-aware score $O_f^k$, the network can still be trained following Eq. 2. During inference, the scores for all sampled frames with intervals of $t$ are calculated, and the final score is obtained by averaging the frame scores, as shown in Eq. 5.

$$Q = (\sum_{j=1}^{T//t} O_f^j)/(T//t), \tag{5}$$

where $Q$ represents the final score of the video predicted by the network, and T represents the total number of frames of the video. In our method, $t$ is set to be 10.

## 3.4 Multi-frequency Feature Fusion

For high-quality 4K videos with richer high-frequency information, existing VQA methods typically feed spatial information directly into the network for training. However, the critical details might be overlooked by the convolution and down-sampling in the shallow layers of DNNs, which leads to less accurate quality predictions for high-quality 4K content.

To address this, multi-frequency feature fusion is proposed to capture sufficient detail information, which utilizes the Haar Transform (HT) to convert 4K videos from the spatial to the frequency domain. A standard HT layer reduces the spatial size of the input while increasing the channel number by a stride-2 convolution with four kernels including $LL^\top, HL^\top, LH^\top, HH^\top$, where $L = \frac{1}{\sqrt{2}}[1, 1]^\top$ and $H = \frac{1}{\sqrt{2}}[1, -1]^\top$. The low-pass filter $LL^\top$ acts as the average pooling on feature maps while the three high-pass filters capture edge-like information with different orientations.

To accurately demonstrate the HT's ability to capture details in 4K videos, we select two 4K videos with similar scenes and extracted frames that are highly similar to each other. By applying HT to these frames and comparing the results, visualizations can be observed in Fig. 4. In this figure, the upper half represents a frame from a higher-scoring 4K video, while the lower half represents a frame from a lower-scoring 4K video. Direct observation of spatial information reveals high quality in both cases. However, upon closer examination of different frequency domains, numerous distinctions become evident, with the higher-scoring video exhibiting richer detail. Therefore, pre-extracting these abundant details in various frequency domains using HT can prevent the loss of such detail information during Linear Embedding. This approach enables the network to more effectively focus on the crucial details essential for 4K VQA.

Specifically, we down-sample the patches half to their original scale through HT. The spatial information of the patches is then represented in the low-frequency component obtained through average pooling, while the high-frequency details from the original scale are preserved in three different edge-like high-frequency maps. Assume that the patches derived from a frame are contained within the set $\mathbb{P}$ and we consider an element $p$ from $\mathbb{P}$ as an example. Suppose the original scale patch $p \in \mathbb{R}^{3 \times l \times l}$, where $l$ represents the height and weight of $p$. Then, the four components resulting from the Haar Transform are $p_{\text{avg}}, p_h^1, p_h^2,$ and $p_h^3$, where $p_{\text{avg}}, p_h^1, p_h^2,$ and $p_h^3 \in \mathbb{R}^{3 \times \frac{l}{2} \times \frac{l}{2}}$. We employ linear embedding layers with shared-weight to transform these multi-frequency maps into embeddings. The network then adaptively fuses these embeddings. The specific process is as follows:

$$p_{\text{avg}}, p_h^1, p_h^2, p_h^3 = \text{Split}(\text{HT}(p)), \tag{6}$$

$$z = \alpha_1 \text{LE}(p_{\text{avg}}) + \alpha_2 \text{LE}(p_h^1) + \alpha_3 \text{LE}(p_h^2) + \alpha_4 \text{LE}(p_h^3), \tag{7}$$

where $z$ represents the input embeddings of the encoder, LE represents the Linear Embedding Layer and $\alpha_i$ represents the learnable parameters. Multi-frequency feature fusion enables the network to

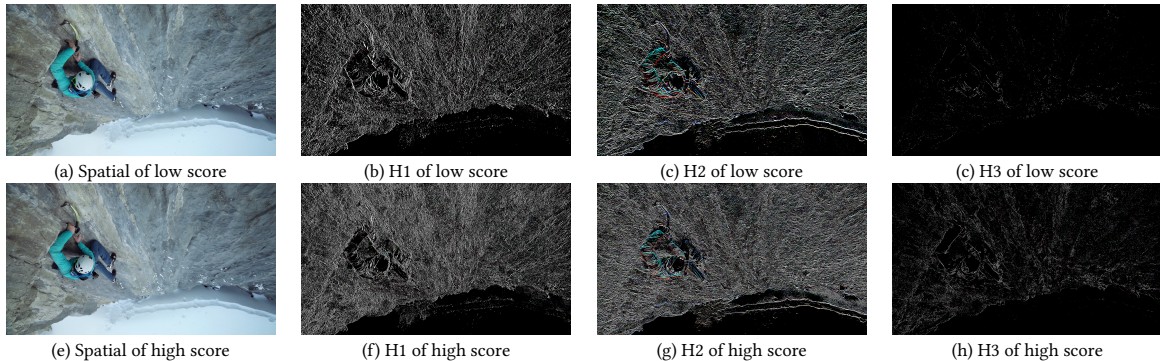

| (a) Spatial of low score | (b) H1 of low score | (c) H2 of low score | (c) H3 of low score |
| (e) Spatial of high score | (f) H1 of high score | (g) H2 of high score | (h) H3 of high score |

Figure 4: Comparing on multi-frequency of two similar frames from different videos.

focus on capturing more detail information regarding edges (extracted from the high-frequency maps). It is noteworthy that, apart from the computation involved in the HT itself, we do not introduce any additional computational cost. Moreover, the input size of the encoder is reduced to 1/4 of that without multi-frequency feature fusion, leading to a significant decrease in computational burden.

## 3.5 4K Video Dataset

*3.5.1 Data collection.* 200 video clips from the digital masters of movies, television dramas, and TV shows are collected from an online video streaming platform, xxxx.com, guided by a criteria that the era, genre, and region of these video clips are distributed as balanced as possible. Firstly, 10 ten-second video sequences are randomly clipped from each of 200 source videos, resulting in a total of 2000 ten-second video clips. Then 200 representative video clips are sampled from the 2000 clips, where the sampling strategy in [39, 44] is adopted. In the sampling process, several video indicators obtained by using [1] are considered including spatial activity, temporal activity, noise, brightness, and contrast. Finally, the distributions of video indicators are shown in Fig. 5, and some video clip samples are illustrated in Fig. 7. It should be noted that these videos are from the digital masters of the video streaming platform, thus, the original resolutions of these videos are ranged from 720P to 4K, with almost the best quality level at the corresponding era/region and the corresponding resolution. In addition, there is also the possibility that the digital master has been post-processed, such as through super-resolution or enhancement, and might have suffered various distortions. All these videos are then up-scaled to 4K for evaluation.

*3.5.2 Subjective Experiment.* In our experiment, the PC method [10] is adopted where the subject is forced to choose which of the two stimuli is preferred. PC is sensitive to the conditions with small differences so that the accurate and reliable subjective opinions for the 4K videos can be obtained. However, the number of comparisons increases exponentially with the number of processed video sequences (PVSs). So we adopt the strategies in [17] to limit the number of comparisons while maximizing the accuracy that can be reached. Specifically, each observer needs to watch 100 pairs of ten-second 4K videos and choose the better one from each pair. The two stimuli are displayed in a side-by-side way on two 4K 27-inches

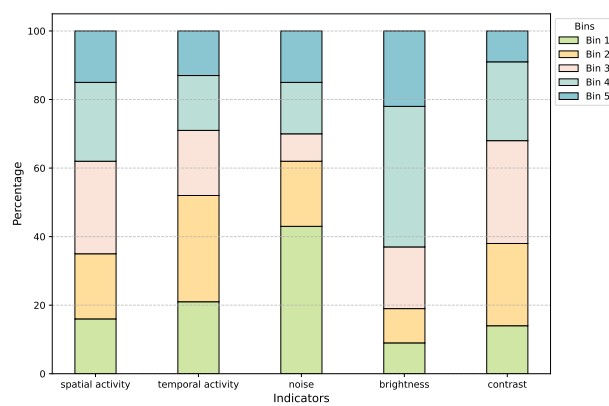

Figure 5: Distribution percentage of video indicators. The normalized feature space of each indicator is uniformly divided into 5 Bins.

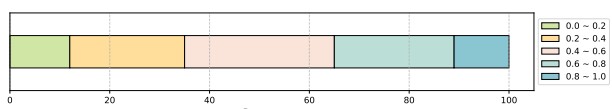

Figure 6: Distribution percentage of MOS in our dataset.

screens (HP-Z27K G3). The viewing distance is approximately 1.6H. All test environment follows ITU-R BT500[33]. The comparison list for the subsequent observer is based on the previous subjective results. On average, each stimulus is observed 100 times in the experiment. The quality scores are re-scaled from 0 to 1. The distribution of the final MOS scores is shown in Fig. 6. Some video clips and their subjective scores are presented in Fig. 7.

## 4 EXPERIMENTS

### 4.1 Experimental Settings

*4.1.1 Datasets.* Experiments are carried out on our developed 4K dataset, which encompasses a diverse range of scenes and quality levels. In addition, two frequently used open-source VQA datasets are included to validate the applicability of our method across different high-resolution VQA tasks. Specifically, one dataset is

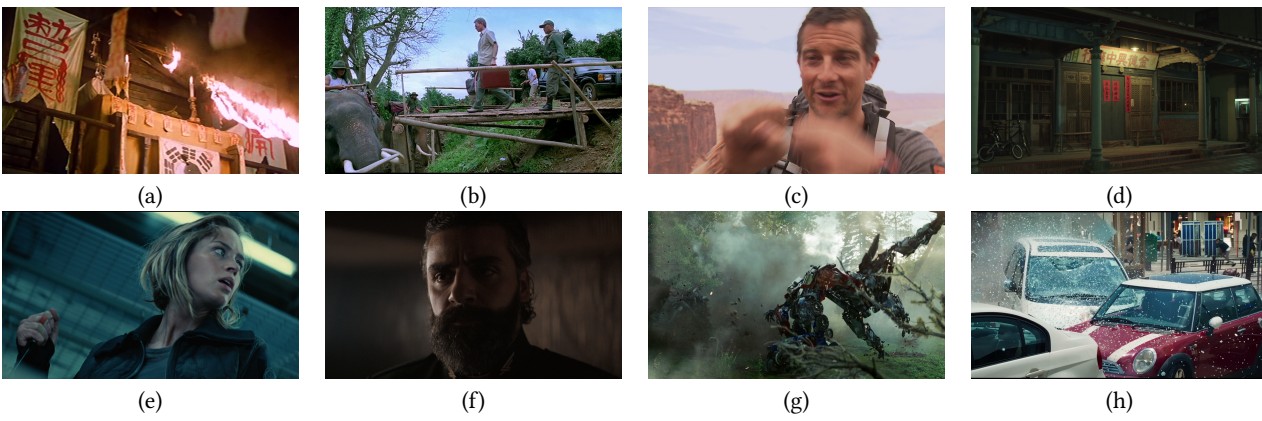

**Figure 7: Frames of video samples in our dataset. Details can be seen more clearly when zoomed in. The MOS scores of the videos that (a)~(b) belong to are 0.08, 0.20, 0.29, 0.41, 0.58, 0.68, 0.72, and 0.81, respectively.**

LIVE-Qualcomm [7] dataset, comprising 1080P videos, while the other is the YouTube-UGC [39] dataset, where 35% of the videos are either 1080P or 4K videos.

*4.1.2 Methods for comparison.* Five deep learning-based methods are used for comparison, including two methods without special design on data sampling including VSFA [8] and BVQA-2022 [14]; three methods with grid-based sampling strategy including FAST-VQA [41], DOVER [43] and SAMA [20].

## 4.2 Performance Criteria

We employ two metrics, including the Pearson Linear Correlation Coefficient (PLCC) and the Spearman Rank-Order Correlation Coefficient (SRCC), to evaluate the performance of VQA. The PLCC computes the linear predictability of the VQA algorithm, and the SRCC assesses the prediction monotonicity. Their values are in the range of [0,1] and the higher value means the better performance.

To guarantee unbiased dataset partitioning and equitable comparisons, we randomly divide the datasets into an 80% training set and a 20% testing set, performing this split 10 times and presenting the median SRCC and PLCC scores. For LiveVQC, LIVE-Qualcomm, and YouTube-UGC datasets, we cite comparison results directly from existing literature when available. For methods not documented in published works, we retrain and evaluate them according to their original optimization protocols.

## 4.3 Performance Comparison

The quantitative comparison results are shown in Table 1. From Table 1, it could be observed that on our dataset specifically constructed for 4K VQA, our method shows significantly better performance over other methods. Specifically, it outperforms the second-best method by 0.063 (an increase of **7.4%**) and 0.050 (an increase of **5.7%**) on SRCC and PLCC, respectively. This proves the effectiveness of our approach in capturing all information of 4K content. On LIVE-Qualcomm, a dataset that exclusively contains 1080P videos, our method also achieves state-of-the-art performance, outperforming the second-best method with improvements of 0.004 and 0.011 on SRCC and PLCC, respectively. This demonstrates our method's

capability on accurately predicting the quality of high-resolution videos that close to 4K resolution. On YouTube-UGC, our method secures second place, scoring 0.011 lower on PLCC than DOVER [43]. A possible reason is that our method is designed for capturing detailed information, and loses its advantage in low-resolution videos, while 42% of videos in YouTube-UGC are 360P or 480P. Overall, the comparison experiments illustrate how our method is effective not just for 4K but also for other high-resolution VQA.

It is worth noting that our method has served in one of the world's most well-known video streaming platforms. It has already provided reliable video quality evaluations in practical business scenarios such as Ultra-HD video QC, digital master QC, video enhancement/super-resolution evaluation, etc.

## 4.4 Ablation Studies

*4.4.1 Ablation Studies for the Proposed Method.* To see the performance contribution of each component within our method, we conduct ablation studies by forming the following baseline models. (a) **"Plain Model I"**: We resize the 4K video frames and input them directly into the network to obtain frame scores. (b) **"Plain Model II"**: We crop patches from the 4K video frames randomly and input them into the network, utilizing the video score to supervise the corresponding patches. (c) **"Plain Model *w/* FuPiC"**: A baseline that only utilizes the proposed FuPiC sampling and training strategy. (d) **"(c) *w/* Region-aware Scheme"**: A baseline that utilizes both the proposed FuPiC strategy and region-aware scoring scheme.

The results of the ablation studies are listed in Table 2, where the final method outperforms all the baselines noticeably. (i) The performance of the plain models (a) and (b) is significantly poor, which highlights that resizing or cropping the video would cause the loss of crucial details or result in the omission of essential visual information. (ii) The result of (c) shows significant improvement compared with (a) and (b), proving the importance of maximizing video information fed into the network. (iii) When comparing (c) with (d), it is evident that not utilizing the region-aware scheme shows a decline in SRCC and PLCC by 0.038 and 0.040, respectively. This indicates that reasonably combining the results of the patches is highly effective. (iv) Against (d), our final method increases SRCC

**Table 1: Performance comparison on four datasets. Bold: best; and Underline: 2nd-best.**

| Method | Source | Data Sampling | LIVE-Qualcomm | | YouTube-UGC | | Our 4K dataset | | *Weighted Average* | |
|---|---|---|---|---|---|---|---|---|---|---|
| | | | SRCC | PLCC | SRCC | PLCC | SRCC | PLCC | SRCC | PLCC |
| VSFA [8] | ACMMM2019 | Traditional | 0.737 | 0.732 | 0.724 | 0.743 | 0.718 | 0.737 | 0.725 | 0.740 |
| BVQA-2022 [14] | TCSVT2022 | Traditional | 0.817 | 0.828 | 0.831 | 0.819 | 0.814 | 0.822 | 0.827 | 0.821 |
| FAST-VQA [41] | ECCV2022 | Grid-based | 0.819 | 0.851 | 0.855 | 0.852 | 0.851 | 0.867 | 0.849 | 0.853 |
| DOVER [43] | ICCV2023 | Grid-based | 0.736 | 0.789 | **0.890** | **0.891** | 0.838 | 0.862 | 0.862 | 0.874 |
| SAMA [20] | AAAI2024 | Grid-based | 0.815 | 0.829 | 0.881 | 0.880 | 0.848 | 0.882 | 0.868 | 0.873 |
| Proposed Method | Proposed | Proposed | **0.823** | **0.862** | 0.876 | 0.883 | **0.914** | **0.932** | **0.873** | **0.886** |

**Table 2: Results of ablation studies for the proposed method.**

| | Method Setting | SRCC | PLCC |
|---|---|---|---|
| (a) | Plain Model I | 0.759 | 0.790 |
| (b) | Plain Model II | 0.819 | 0.824 |
| (c) | Plain Model *w/* FuPiC | 0.834 | 0.853 |
| (d) | (c) *w/* Region-aware Scheme | 0.872 | 0.893 |
| (e) | Proposed Method | 0.914 | 0.932 |

and PLCC by 0.042 and 0.039, respectively. This suggests that effectively capturing the details of 4K videos from the frequency domain enables the network to assess video quality more accurately. Moreover, we conduct an experiment on replacing the shared linear projection with distinct linear projection, the results decrease by 0.035 and 0.027 on SRCC and PLCC, respectively, suggesting that there is no need to utilize different linear projection specifically designed for different frequency information. Instead, a simple projection is enough.

**Table 3: Results of ablation studies for non-local/position information.**

| Method Setting | SRCC | PLCC |
|---|---|---|
| *w/* Non-local Information | 0.914 | 0.930 |
| *w/* Positional Information | 0.906 | 0.918 |
| Proposed Method | 0.914 | 0.932 |

*4.4.2 Ablation Studies for Non-local/Position Information.* In addition to the previous experiments, we set up two more baselines to study whether the incorporation of non-local or positional information is useful to our proposed method or not. (f) **"*w/* Non-local Information"**: We integrate embeddings containing information from neighboring regions with those of the current patch, feeding these combined features into the network. (g) **"*w/* Positional Information"**: We introduce learnable position encodings based on the spatial locations of the patches. These encodings are also merged with the embeddings of the current patch before being fed into the network.

The results presented in Table 3 indicate that fully leveraging the detailed information in 4K videos and reasonably combining the results from different regions is sufficient. Adding non-local/positional information would increase computational cost and even lead to a decline in performance.

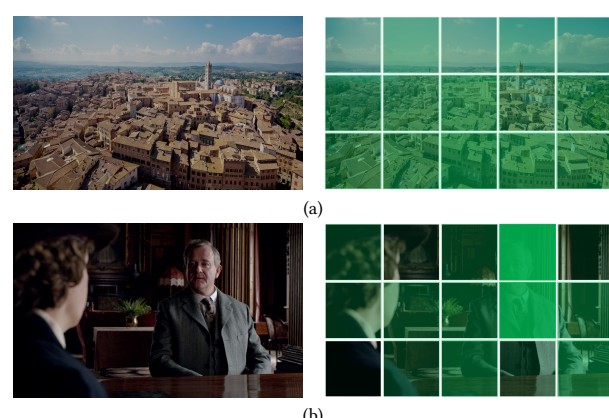

(a)

(b)

**Figure 8: Visualization of the region-aware scheme. The right side shows the visualization of the final weight for each patch, with darker green indicating higher importance.**

## 4.5 Visualization on Region-aware Scoring Scheme

In addition to comparative results, The visualization of the weights for the regions in 4K frame predicted by our Region-aware Scoring Scheme is illustrated in Fig. 8. The figure shows that the network effectively focuses on the relatively important regions, which proves highly beneficial for the 4K VQA task.

## 5 CONCLUSION

A novel and highly efficient NR 4K VQA method is presented in the paper. The main component of this method is the proposed FuPiC sampling and training strategy, which can feed all the original content of 4K frames into our VQA network under computational resource limitations. A region-aware scoring scheme is designed to mimic human subjective perception, where each sub-region contributes differently to the overall score of the 4K frame. Additionally, a novel multi-frequency feature fusion approach is developed to enhance the performance of the proposed method and significantly reduce inference time. Experiments prove the effectiveness of the proposed method on our constructed 4K video quality dataset and other open-source video quality datasets, resulting in high practicability.

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
