# OpenReview forum: "Highly Efficient No-reference 4K Video Quality Assessment with Full-Pixel Covering Sampling and Training Strategy"
_acmmm.org/ACMMM/2024/Conference — MM2024 Poster_

### Official Review · Reviewer_H2Yh · 2024-05-07

**Rating:** 2
**Confidence:** 4

**Summary:**

The paper presents a dataset and a model for 4K video quality assessment.
for me it is in the beginning unclear which exact part of VQA is this paper aiming, user-generated content, short term video or long term video, video compression.
The overall approach looks fine, and interessting, however, it is not mentioned that the data or the model will be published, so I see only a minor effect to the research community.
Well, I agree that the combination of some aspects of the model is new, and the dataset is larger than other datasets, however if not shared it is limited.
And also the ideas are not as "new" as indicated in the paper.
Especially the lack of SoA and defining which aspect of VQA is aimed in this paper makes it hard to accept.

# comments

## abstract
* "To our knowledge, this is the first technology for the NR 4K VQA task.", well a) there are FFT based video quality features available (e.g. to measure blurriness, blockiness), b) if this is referred to "using DNNs to process full 4K video frames" it's also not true, e.g. DeViQ (https://library.imaging.org/ei/articles/30/14/art00017) processes full-4k frames for video quality prediction. So I would propose to not at such a "to our knowledge" statement

## introduction
* what about P.1204 family of video quality models? there is e.g. a no-ref model P.1204.3 and a reduced-ref model P.1203.4 and a hybrid model P.1204.5
* "In subjective 4K video assessments, humans evaluate various regions, such as focal areas, background, and blur-affected regions due to
jitter, each impacting overall quality differently. Thus, we consider partitioned patches as a series of regions, using a neural network to learn their weights and scores to estimate overall video quality", any reference for this? what about saliency models? there was work considering saliency models to improve encoding efficiency.

## related work

* 2.1. there are far more SoA classical models available, e.g. https://ieeexplore.ieee.org/document/9846967 , https://ieeexplore.ieee.org/document/9355144 , VMAF? or the mentioned standardized models?
* is it for short term, or long term video quality?
* so what is the aim for the dataset and the model, technical quality with encoding parameters or visual "aesthetic" quality?

* 2.3: datasets for 4K VQA:
    * what about (just a small selection of google scholar search):
        * https://alexandrosstergiou.github.io/datasets/Inter4K/index.html
        * https://media.withyoutube.com/
        * https://ieeexplore.ieee.org/stamp/stamp.jsp?arnumber=10337713
        * https://ieeexplore.ieee.org/document/10178493
        * https://ieeexplore.ieee.org/document/6603201
        * https://ieeexplore.ieee.org/stamp/stamp.jsp?arnumber=8959059
        * https://dl.acm.org/doi/pdf/10.1145/3339825.3394937
        * https://www.sciencedirect.com/science/article/pii/S1047320321002479
        * https://mcml.yonsei.ac.kr/downloads/4kuhdvideoquality
* "In contrast, our constructed 4K dataset
is built on 200 4K videos with a diverse range of scenes and quality levels" technical quality or aesthetics? it should be clarrified what is the purpose of the dataset and which aspect of VQA is handled.
* "Only a few methods [23, 47] have been  specifically designed for 4K image/video quality assessment tasks due to the unique challenges posed by high-resolution content" well please check SoA again, there are more publications for this, even for higher res. images.

## proposed method
* Fig 3 is not needed, it is clear that downscaling vs patching removes details.
* "patches(training samples)" space before the ( missing
* Eq. 1: not needed, it is an average
* Eg. 2: not needed its the definition of MSE
* Eq. 5: T//t a integer division in python (for this there would be a more agreed on math notation)? or a mistake?

* 3.5
    * xxxx.com  youtube? vimeo? can the videos be shared later, will the dataset be shared? can it be reproduced?
* ITU-R BT500 or ITU-R BT500/15?
* Figure 6, well a distribution plot would make this plot easier to process. same holds for fig 5.
* 3.5.2 Subjective Experiment.
    * was there now compression applied?
    * why Paired Comparsion.
    * why two screens? with 27 inches, with this specific distance it may be hard to compare?
        * checkout other approaches for quality evaliuation, such as stripes/temporal switching
## experiments
* why now the LIVE dataset with 1080p resolution, outlines above are more 4K datasets.
* were the comparison models also retrained?
* "This proves the effectiveness of our approach in capturing all information of 4K content", an improve in the 2nd decimal of the metrics... maybe not so "effective", other statements are considering the 3rd decimal ... I would not say it is a prove of effectiveness. yes it is better, but slightly, is it significant? not sure. the comparison models are not retrained, but the reported values for the model are mean across folds? not sure.
* why is Q-Align not used in the evaluation?
* 4.5 so you use a weighted approach of the patches?, how much impact have the individual patches?, how much faster it is to just use a few patches?
* fig 8. what is darker green meaning? bottom left patch ? dark green, vs 4th patch top, light green? or is it the opposite?

## conclusion
* "under computational resource limitations" which? was there an evaluation with this? It was not even mentioned how much GPU memory or so is needed?

**Strengths:**

* kind of new NR model and dataset for 4K video

**Limitations:**

* introduction could be shortened by 1/3, and instead the SoA could be extended, SoA is technically only 75% of a page which is far too less for a paper in the VQA domain.
* taking the full frame not resized with patches is not as new as it is portrait in this paper.
* SoA, and also some statements that state "it is super new" what is done, e.g. only a few datasets for 4K videos,
* having the model and data available would be nice for the community, but unclear
* I pointed out some missing/additional references (not meant that all of them must be included, however they should be considered, e.g. especially the datasets and model references), or other works in the field, which are missing.
* visualization, e.g. Fig 5, and 6, could be done better.
* "We have constructed the first dataset explicitly tailored for NR 4K VQA", sure? there are other datsets with 4K available as pointed out before
* emphasizing the computational resources, but not showing a comparison, I don't know if this model is now a) faster, b) requires less GPU power/memory than other models, it's not shown.

**Suitability:**

2

---

### Official Review · Reviewer_JgDr · 2024-05-26

**Rating:** 4
**Confidence:** 3

**Summary:**

This paper proposes FuPiC, a revision of sampling strategy for video quality assessment that includes following steps: (1) divide videos into large patches that cover all pixels; (2) aggregate the scores from each patch via a learnable module. The authors also propose a frequency-domain sampler to further improve the inputs. The resulting method is claimed to be more effective on 4K video QA datasets.

**Strengths:**

1. The paper is well-motivated.
2. The paper proposes a 4K video dataset to validate its approach.

**Limitations:**

1. The size of the 4K dataset (200) is not big enough.
2. The method may cause high computational load. Can the authors evaluate on that?
3. Why is the frequency part especially important for FuPiC? Do they have any intrinsic connection?

**Suitability:**

2

---

### Official Review · Reviewer_zj1p · 2024-06-09

**Rating:** 4
**Confidence:** 2

**Summary:**

In this paper, Author proposed a video quality assessment method in no-reference settings for 4K videos.

Contributions includes:
>> Full-Pixel Covering for video quality assessment.
>> Weighting and Scoring for Full-Pixel Covering.
>> Dataset curated for non-reference 4k  video quality assessment.

**Strengths:**

>> Paper is well-written and easy to understand.
>> Proposed method outperforms existing video quality assessment methods on multiple datasets.
>> Sufficient ablation studies have been conducted to validate the claims.

**Limitations:**

[Minor]
>> Please highlight some potential applications of this approach.
>> Figure 2 needs to be more detailed.
>> rewrite line 375 "The grey area that outputs the score for each frame represents FuPiC".
>> Line 160 is little unclear, " Generally, 4K videos are partitioned into non-overlapping patches in our study, ensuring the content can be processed within a single GPU for training." Please explain why non-overlapping patches and single GPU training are related
>>In Table 1, use up and down arrows with the parameters names to indicate whether a higher value or a lower value is preferable.

**Suitability:**

2

---

### Meta-Review · Area_Chair_AqUS · 2024-07-07

**Recommendation:** Accept (Poster)
**Confidence:** 2

**Metareview:**

The paper obtained 2 "borderline accept" and 1 "weak reject" in the first round of reviews. Only the reviewer who provided the "weak reject", which was the most confident one and the one that provided the most deep review, commented and updated the review after the rebuttal, leaning towards acceptance.
Thus, I would recommend the acceptance of the paper given that in seemed to provide useful insights, but I recommend the authors to consider the comments carefully and improve the camera-ready version.